# IGF2 May Enhance Placental Fatty Acid Metabolism by Regulating Expression of Fatty Acid Carriers in the Growth of Fetus and Placenta during Late Pregnancy in Pigs

**DOI:** 10.3390/genes14040872

**Published:** 2023-04-05

**Authors:** Zhimin Wu, Guangling Hu, Yiyu Zhang, Zheng Ao

**Affiliations:** 1Key Laboratory of Animal Genetics, Breeding and Reproduction in the Plateau Mountainous Region, Ministry of Education, College of Animal Science, Guizhou University, Guiyang 550025, China; 2Guizhou Provincial Key Laboratory of Animal Genetics, Breeding and Reproduction, College of Animal Science, Guizhou University, Guiyang 550025, China

**Keywords:** pig, placenta, fatty acid carriers, IGF2, DNA methylation

## Abstract

Fatty acids (FAs) are essential substances for the growth and development of the fetus and placenta. The growing fetus and placenta must obtain adequate FAs received from the maternal circulation and facilitated by various placental FA carriers, including FA transport proteins (FATPs), FA translocase (FAT/CD36), and cytoplasmic FA binding proteins (FABPs). Placental nutrition transport was regulated by imprinted genes *H19* and insulin-like growth factor 2 (*IGF2*). Nevertheless, the relationship between the expression patterns of H19/IGF2 and placental fatty acid metabolism throughout pig pregnancy remains poorly studied and unclear. We investigated the placental fatty acid profile, expression patterns of FA carriers, and H19/IGF2 in the placentae on Days 40 (D40), 65 (D65), and 95 (D95) of pregnancy. The results showed that the width of the placental folds and the number of trophoblast cells of D65 placentae were significantly increased than those of D40 placentae. Several important long-chain FAs (LCFAs), including oleic acid, linoleic acid, arachidonatic acid, eicosapentaenoic acid, and docosatetraenoic acid, in the pig placenta showed dramatically increased levels throughout pregnancy. The pig placenta possessed higher expression levels of CD36, FATP4, and FABP5 compared with other FA carriers, and their expression levels had significantly upregulated 2.8-, 5.6-, and 12.0-fold from D40 to D95, respectively. The transcription level of *IGF2* was dramatically upregulated and there were corresponding lower DNA methylation levels in the *IGF2* DMR2 in D95 placentae relative to D65 placentae. Moreover, in vitro experimentation revealed that the overexpression of IGF2 resulted in a significant increase in fatty acid uptake and expression levels of *CD36*, *FATP4,* and *FABP5* in PTr2 cells. In conclusion, our results indicate that CD36, FATP4, and FABP5 may be important regulators that enhance the transport of LCFAs in the pig placenta and that IGF2 may be involved in FA metabolism by affecting the FA carriers expression to support the growth of the fetus and placenta during late pregnancy in pigs.

## 1. Introduction

The placenta is a temporary organ mediating maternal–fetal nutrient and oxygen exchange, synthesizing a great deal of signaling factors and hormones that act to adapt maternal physiology, metabolism, and behavior to support fetal growth and sustain pregnancy [1,2]. In pigs, the chorion and allantois of the placenta are fused by Day 40 (D40) of gestation, and the placenta develops completely in terms of weight, surface area, and numbers of areolae by Day 65 (D65) of gestation [3]; in addition, the placental folds have expanded by Day 95 (D95) of gestation [4]. Hence, placental growth and the development of the placenta require a constant supply of nutrients; consequently, there may be significant differences in fatty acid (FA) needs at different stages of pregnancy.

In term of basic nutrients, FAs are essential substances required by the placenta, given that they serve as a source of energy, are an important component of membranes, and are precursors of various bioactive compounds, such as prostaglandins, prostacyclins, thromboxanes, and leukotriene [5]. Therefore, regulation of the FA metabolism by maternal plasma is crucial for the growth and development of both the placenta and the fetus [5]. FA metabolism in the placenta is a complex process that involves several proteins [6]. Although the passive diffusion and FA transporters both contribute to FAs crossing the placenta from maternal circulation to the fetus, many studies revealed that placental uptake and the transfer of the bulk of these FAs are mainly determined by FA transporters [7,8]. FA transport proteins (FATPs), FA translocase (FAT/CD36), and cytoplasmic FA binding proteins (FABPs) are the major FA carriers, and were proposed as regulators of FA uptake in human placental membranes [9,10]. The collaboration of the above FA carriers ensures that the fetus receives sufficient FAs from the dam across the placental barrier. FAs are essential for the proliferation of trophoblast cells and vascularization [11], which contribute to the morphologies and functions of the placenta; this may be associated with the spatiotemporal expression of FA carriers throughout pregnancy.

Imprinted genes play a crucial role in embryonic growth and development, as well as in placental function, especially the two most widely-researched imprinted genes: H19 and insulin-like growth factor 2 (IGF2) [1,12]. H19/IGF2 are controlled by the differentially methylated region (DMR): methylation in the DMR affects the binding of CTCF (CCTCC-binding factor), controlling their differential expression [13]. Previous studies have also shown that H19/IGF2 exert their influence upon fetal resources from maternal circulation by regulating placental morphology and directly regulating nutrient transport signaling pathways [14,15,16]. Nevertheless, the relationship between the expression patterns of H19/IGF2 and placental fatty acid metabolism throughout pig pregnancy remains poorly studied and unclear. Thus, the present study aimed to determine associations of placental fatty acid metabolism with the expression of FA carriers and H19/IGF2 in D40, D65, and D95 pig placentae. 

## 2. Materials and Methods

### 2.1. Tissue Collection

All experimental protocols involving the use of pigs in this study were approved by the Institutional Animal Care and Use Committee, Guizhou University, Guiyang, China (Animal protocol approval number: EAE-GZU-2020-T010). A total of 20 Duroc sows were artificially inseminated thrice (8, 20, and 32 h) after showing signs of standing estrus with the semen collected from the same Duroc boar. All sows were raised on the same farm under the same conditions. The pregnant sows were hysterectomized after the induction of anesthesia on D40, D65, and D95 of pregnancy (n = 5 sows/day of pregnancy) and the uteri were then rapidly transported in an icebox to the laboratory. The remaining five sows were allowed to farrow piglets via spontaneous labor. The uteri were opened at the horns at the antimesometrial side of the uterus. Placental samples from each fetus were collected at the implantation site, immediately snap-frozen in liquid nitrogen, and stored at −80 °C. However, for the following analyses, only the placental samples of two fetuses with body weights similar to the mean litter weight in one litter were employed. For the fatty acids profiles and qPCR tests, eight placenta samples from each gestational stage were used. The same samples were placed in containers and fixed in 4% paraformaldehyde, to be used for histomorphological and immunofluorescence examination. Three placenta samples were randomly chosen from the above eight placenta samples used for Western blotting and bisulfite sequencing analyses.

### 2.2. Determination of FA Profiles

The FAs of the placenta tissue samples (50 mg) were extracted with a chloroform–methanol (2:1 *v*/*v*) mixture in accordance with the method as described by a previous study [17]. The extracted FAs were transesterified with a 2 mL 1% methanol–sulfuric acid mixture for esterification for 30 min at 80 °C. The 5 mL H_2_O (4 °C) and 1 mL hexane were added, vortex mixed, and centrifuged for 10 min, 4 °C, 12,000 rpm. Methyl ester derivatives in the organic layer added with 15 µL internal standard (i.e., methyl salicylate; TCI, Tokyo, Japan) were tested using a ThermoFisher Trace 1310 ISQ gas chromatograph mass spectrometer with a TG-5 mass spectrometry (MS) capillary column (50 m × 0.25 mm × 0.20 µm) and were carried by He gas. The sample injection volume was 1 µL and the flow rate was 0.63 mL/min. The following values were fixed for the electrospray source parameters: initiation and holding at 80 °C for 1 min; heating to 160° C at 20 °C·min^−1^ for 1.5 min; heating to 196 °C at 3 °C·min^−1^ for 8.5 min; heating to 250 °C at 20 °C·min^−1^ for 3 min. The temperature of the ion source was set at 230 °C; Supelco 51 component FA methyl ester (FAME) mix (Nu-Chek Prep, Inc., Elysian, MN, USA) was used as the reference standard to identify FAs. The Electron Bombardment Ionization (EI) source and SIM scan mode with an electron energy 70 eV were used for MS detection. Authentic standards (Thermo Fisher, Waltham, MA, USA) were used to identify FAMEs. Based on the peak regions of the associated FAMEs, the relative contents of the membrane FAs were measured. The quantitative contents of the FAs were then converted into percentages of the total FAs.

### 2.3. Quantitative Real-Time Polymerase Chain Reaction (PCR)

The total RNA was extracted from placental tissues using a Total RNA Kit (OMEGA, Norcross, GA, USA) in accordance with the manufacturer’s protocol. Subsequently, cDNA was synthesized from 1 µg total RNA using a PrimeScript RT Reagent Kit (TakaRa, Shiga, Japan). Quantitative gene expression analysis was performed using a SYBR Select Master Mix Kit (Thermo Fisher Scientific, Waltham, MA, USA) following the manufacturer’s instructions. Table 1 presents the primer information for target genes. The reactions were carried out by a QuantStudio™ 3 Flex Real-Time PCR System (Thermo Fisher Scientific, Waltham, MA, USA). The reactions were carried out in the following ways: initial denaturation at 95 °C for 2 min, followed by 40 cycles of denaturation at 95 °C for 15 s, annealing at 60 °C for 15 s and an extension at 72 °C for 1 min, and a final melting cycle made up of 95 °C for 15 s, 60 °C for 1 min, and 95 °C for 15 s. All gene expression experiments were carried out in triplicate. β-Actin was used as the house-keeping gene, and the 2^−ΔΔCt^ method was used to determine the relative expression levels.

### 2.4. Western Blotting Analysis

Western blot analyses were carried out as described previously [17]. The total protein of the placenta samples was extracted using a Tissue Protein Extraction Kit (CWBIO, Jiangsu, China), and the concentration of the extracted total protein was determined by a bicinchoninic acid assay (CWBIO, Jiangsu, China). Sodium dodecyl sulphate–polyacrylamide gel electrophoresis and transmembrane experiments were carried out with the Mini-PROTEAN Tube Cell instrument (Bio-Rad, Hercules, CA, USA). The membranes were incubated with primary antibodies, including anti-FATP4 (ProteinTech, Rosemont, IL, USA), anti-CD36 (ProteinTech, Rosemont, IL, USA), anti-FABP5 (ProteinTech, Rosemont, IL, USA), and anti-β-actin primary antibodies (Cell Signaling Technology, Danvers, MA, USA). Thereafter, the membranes were incubated with horseradish peroxidase-conjugated goat anti-rabbit secondary antibody (Santa Cruz company, Hercules, CA, USA), and immunoreactive bands were visualized using the ECL kit (Beyotime, Shanghai, China) and documented with a chemiluminescence imaging system (UVP, Upland, CA, USA). The results of the densitometric analysis of bands were quantified by Image J 1.45 software (National Institutes of Health, Bethesda, MD, USA). To calculate the relative expression levels of the target proteins, all blots were standardized to β-actin.

### 2.5. Histomorphological Analysis

Following the routine histological process, placental tissues were fixed for 24 h in buffered 4% paraformaldehyde, dehydrated in increasing amounts of ethanol, and cleared with xylene. They were then embedded in paraffin and sectioned at a thickness of 5 μm. The sections were dewaxed, rehydrated, and stained with hematoxylin and eosin following standard histological procedures. In this study, morphometric measurements of the stained sections were performed as described previously [19]. To obtain the average width of the PFs within each measured field, a photomicrograph of each stained section of the top and bottom of the placenta folds (PFs) was obtained using a Nikon 80i light microscope (100× magnification) fitted with a Nikon (DS-Fi1) digital camera (Nikon, Tokyo, Japan). The obtained images were marked and these points were joined to form a polygon. Individual lines were then drawn from the top to the bottom of each fold to obtain the central point of each fold. Then, a connecting line (L) was drawn through the center of each fold within the polygon. The area of the polygon, the length of the L, and the number of trophoblast cells within the polygon were calculated using Image J 1.45 software (National Institutes of Health, Bethesda, MD, USA). Thus, the average width of the PFs was calculated as the polygon area divided by L. The number of trophoblast cells within the polygon was divided by the area of the polygon to obtain the number of trophoblast cells per unit area of placenta. Three different fields were used for the morphometric measurements for each section.

### 2.6. Bisulfite Sequencing Analysis

Genomic DNA was extracted from placental samples using a Total DNA Kit (OMEGA, Norcross, GA, USA). An EZ DNA Methylation-Gold Kit (Zymo Research, Irvine, CA, USA) was used for bisulfite modification of genomic DNA samples, following the manufacturer’s protocol, and according to the protocol of a previous study [18]. Briefly, the DNA was treated with bisulfite followed by the conversion of unmethylated cytosine to uracil. Nested PCR amplifications of bisulfite-treated DNA were performed using the primers in Appendix A, as described previously [18]. PCR amplification was carried out using Hot Start Taq™ Polymerase (Zymo Research, Irvine, CA, USA) in a 10 mL reaction volume with a thermo profile of 98 °C for 5 min, 94 °C for 30 s, 72 °C for 30 s, for 40 cycles, followed by 72 °C for 10 min. The annealing temperature used for each primer pair is shown in Appendix A. The products from the first amplification reaction were used as templates for the second PCR reaction. The amplified products were purified using a DNA Purification Kit (OMEGA, Norcross, GA, USA). Purified fragments were cloned into the pMD19-T Vectors (TakaRa, Shiga, Japan). PCR amplification products from all samples were sequenced and later analyzed using the BiQ Analyzer software.

### 2.7. Immunofluorescence Analysis

Immunofluorescence (IF) was performed by standard IF procedures as previously described [20]. In brief, paraffin-embedded sections were cleared in xylene and rehydrated in an alcohol gradient. Fixed samples of the term placenta were sectioned at 4 µm thickness. The sections were repaired with citric acid by soaking in an antigen retrieval solution using boiling water at 100 °C for 10 min. Subsequently, the sections were incubated in 3% hydrogen peroxide (H_2_O_2_) for 10 min at room temperature, washed thrice with phosphate-buffered saline (PBS), and were then blocked with 2% bovine serum albumin in PBS for 1 h. Next, the sections were incubated with rabbit FATP4, FABP5, CD36, and β-actin antibodies (identical to Western blot) at 4 °C overnight in humid chambers. Secondary-antibody goat anti-rabbit immunoglobulin G (IgG, Thermo Fisher Scientific, Waltham, MA, USA) was incubated with the sections for 1 h. The sections incubated with normal mouse IgG (Servicebio, Wuhan, China; GB111739) substituted for primary antibodies (applied at the same concentration as was used for the specific antibody) were used as negative controls as descried previously [21]. A photomicrograph of each IF-stained section was obtained using a Nikon 80i light microscope (×200 magnification) fitted with a Nikon (DS–Fi1) digital camera (Nikon, Tokyo, Japan). The intensities of the fluorescent signal of all proteins were measured with the Image J 1.45 software (National Institutes of Health, Bethesda, MD, USA). Four different fields were used for the measurements of each section. The relative fluorescence intensities of FATP4, FABP5, and CD36 were calculated by dividing their intensities by β-actin intensity. All sections were stained under the same conditions.

### 2.8. Construction of IGF2 Overexpression Plasmid

The coding sequences (CDS) of the porcine IGF2 gene were obtained from the NCBI database (https://www.ncbi.nlm.nih.gov/nuccore/NM_213883.2) (accessed on 11 May 2022). The IGF2 CDS was analyzed to identify appropriate restriction enzymes for vector construction using Primer 5.0 software (PRIMER-E Ltd., Plymouth, UK). The recombinant plasmid containing the IGF2 CDS and the pEGFP-C1 plasmid were subjected to digestion with *HindII*I and *Kpn*I, and the linked product was subsequently transformed into DH5α competent cells. Restriction enzyme digestion and sequencing were used to identify endotoxin-free plasmids containing the correct fragments.

### 2.9. Cell Culture and Transfection

The porcine trophoblast cell line PTr2 was kindly provided by South China Agricultural University as previously described [22]. The PTr2 cells were cultured in DMEM-F12 (Gibco, Carlsbad, CA, USA) supplemented with 10% fetal bovine serum (Gibco, Carlsbad, CA, USA) at 37 °C and 5% CO_2_ and recombinant human insulin (Yeasen, Shanghai, China). They were then transferred into a six-well plate with 0.25–1 × 10^6^ cells per well. The pEGFP-C1- IGF2 and pEGFP-C1 plasmid were transfected into cells using Lipo8000 (Beyotime, Shanghai, China) in accordance with the manufacturer’s protocol.

### 2.10. Fatty Acid Uptake Assay

Fluorescent BODIPY fatty acids are very long-chain fatty acid analogues and are frequently used to measure cellular fatty acid uptake. In brief, PTr2 cells were cultured on the round coverslip placed in a 48-well plate containing 300µL complete medium (10% FBS, 0.1% insulin. IGF2-Vector and NC-Vector were transfected with Lipo8000 (Beyotime, Shanghai, China) after the cells reached 80% according to the manufacturer’s protocol. The culture was washed twice with PBS after 24 h, and 2µM BODIPY FL C16 (ThermoFisher Scientific; Waltham, MA, USA), a fluorescent palmitate analogue 4, 4-difluoro-5, 7-dimethyl-4-boro-3a, 4a-diaza-s-indacene-3-hexadecanoic acid, which undergoes native-like transport and metabolism in cells, and 5% BSA (Solarbio, Beijing, China) without fatty acids, were added to each well. After mixing, the cells were incubated at 37 °C in a 5% CO_2_ incubator for 1 min, 5 min, 15 min, and 30 min, respectively. Each time point has six replicates. After incubation, the round coverslip was washed twice with PBS and fixed for 30 min with 4% paraformaldehyde (Leagene, Shanghai, China). After washing twice with PBS, the round coverslip with fixed cells was picked up with tweezers and placed on slides; a confocal microscope (Nikon-Eclipse-Ti, Tokyo, Japan) was used to quantify the uptake of fatty acids.

### 2.11. Statistical Analysis

Least squares regression analyses were used to test the effects of the day of gestation for the width of placental folds, the number of trophoblast cell per unit area of placenta, placental FA concentrations, mRNA, and protein expression levels across the different gestation periods. Differences in these variables were analyzed using one-way analysis of variance in the SPSS 19.0 software (IBM Corp., Armonk, NY, USA). Significant differences in the means of two different groups were determined at *p* < 0.05.

## 3. Results

### 3.1. The Trajectory of Fetal Weight and Placental Morphologies in Pig Pregnancy

In this study, three sows miscarried; as a result, the following were obtained: 45 fetuses derived from four sows at Day 40 of gestation (D40), 41 fetuses derived from four sows at Day 65 of gestation (D65), 49 fetuses derived from four sows at Day 95 of gestation (D95), and 45 piglets delivered by five sows. Weight gain in the pig fetuses accelerated and was mainly concentrated during late gestation (Appendix A). A visual assessment of the maternal–fetal interface sections stained with hematoxylin and eosin revealed differences in the morphological characteristics of D40, D65, and D95 placentae: the morphologies of the D65 placentae tended to be complete at Day 65 of gestation (Figure 1A); in particular, the number of trophoblast cells was increased significantly than that of D40 placentae (Figure 1C), and the width of placental folds dramatically increased from Days 40 to 95 of gestation (*p* < 0.01) (Figure 1B).

### 3.2. Comparison of FA Concentrations in Porcine Placentae on D40, D65, D95, and Term Pregnancy

The percentage of total lipid plasma FAs was used to represent the FAs’ composition. A total of 49 FAs were detected and the mean FA percentages in the placentae in the four gestation periods are shown in Table 2. The levels of myristelaidic acid (C14:1T), myristoleic acid (C14:1), and 10-Heptadecanoic acid (C17:1) in the placentae showed significantly decreased (*p* < 0.05) trends from D40 to D95 of gestation. Nevertheless, the concentrations of oleic acid (C18:1N9C), vaccenic acid (C18:1N7), linoleic acid (C18:2N6), behenic acid (C22:0), cis-8,11,14-Eicosadienoate acid (C20:3N6), arachidonic acid (C20:4N6), eicosapentaenoic acid (C20:5N3), lignoceric acid (C24:0), docosapentaenoic acid (n-3) (C22:5N3), and docosahexaenoic acid (C22:6N3) were dramatically increased in D95 placentae compared with those in D40 placentae (*p* < 0.05) (Table 2).

### 3.3. Expression Levels of FA Transport-Related Genes in Porcine Placentae on D40, D65, D95, and Term Pregnancy

In this study, we compared the mRNA expression levels of *FATP1*, *FATP2*, *FATP3*, *FATP4*, *FABP3*, *FABP5*, *FABP7*, *CD36*, and *LPL* genes in D40, D65, D95, and full-term placentae. Among these genes, *FATP1*, *FATP2*, *FATP3*, *FABP3*, *FABP7*, and *LPL* had extremely low transcription levels in D40, D65, D95, and full-term placentae (Figure 2). The placental expressions of *CD36*, *FATP4*, *FABP5*, *FABP3*, and *LPL* mRNA in D65 placentae were higher (*p* < 0.05) than those in D40 placentae (Figure 2). *FATP4* and *CD36* had significantly increased expression levels in D95 placentae compared with those in D40 and D65 placentae (*p* < 0.05) (Figure 2). The expression level of *FABP5* was dramatically upregulated in D95 placentae compare with that in D40 placentae, but did not differ from that in D65 placentae. Additionally, the mRNA abundances of *FATP4*, *CD36*, and *FATP2* were significantly lower (*p* < 0.05) in full-term placentae than those in D95 placentae (Figure 2). However, there was no difference in the mRNA expression levels of *FATP1*, *FATP3*, and *FABP7* in D40, D65, D95, and full-term placentae (*p* > 0.05) (Figure 2). 

Next, the protein expression levels of CD36, FATP4, and FABP5 were compared across D40, D65, D95, and full-term placentae. The placental protein expressions of CD36, FATP4, and FABP5 were significantly upregulated 2.8-, 5.6-, and 12.0-fold from D40 to D95 of gestation (*p* < 0.05), respectively (Figure 3), which was consistent with their transcription levels (Figure 3). Nevertheless, we observed no difference in the protein expression levels of CD36, FATP4, and FABP5 (*p* > 0.05) between D95 and full-term placentae (Figure 3).

### 3.4. Localization and Expression of FATP4, FABP5, and CD36 in Pig Placentae

To explore the localization and expression of the main placental FA carriers during late pregnancy, FATP4, FABP5, and CD36 proteins were immunodetected in pig full-term placentae. The fluorescent signals of FATP4, FABP5, and CD36 were detected in the cytotrophoblastic layer of placental terminal villi (Figure 4A). We also observed similar intensities of fluorescent signals of FATP4, FABP5, and CD36 (Figure 4B).

### 3.5. Imprinting Status of IGF2/H19 Was Altered in Porcine Placentae with Gestation Progress

As shown in Figure 5A, the expression level of *IGF2* in D95 placentae was significantly higher than that in D65 placentae (*p* < 0.05), but there was no significant difference between D95 and D40 placentae. There was no significant difference in the transcription level of *H19* in the three gestation periods (*p* > 0.05), but there was an increasing trend in the placenta of D65 (Figure 5B). Subsequently, we detected the DNA methylation levels of *IGF2* DMR2 and *H19* CTCF3 in porcine placentae. Based on the primers of a previous study [18], DMR2 and CTCF3 contained 24 and 12 CpG sites, respectively. The results show that the DMR2 in the D65 placentae was highly methylated compared with that in the D40 and D95 placentae (Figure 5C). The methylation levels of CTCF3 in the D65 placentae was lower than that in the D40 and D95 placentae, but the difference was not significant (*p* > 0.05) (Figure 5D). 

### 3.6. Overexpression of IGF2 Enhances Fatty Acid Uptake by Porcine Trophoblast Cells

To investigate the effect of overexpression of IGF2 on fatty acid uptake in PTr2 cells, we expressed IGF2 transiently and incubated the cells with BODIPY FL C16. A fluorescent Bodipy fatty acid assay revealed a significant increase in the uptake of fluorescent fatty acids when PTr2 cells were transfected with IGF2 overexpression vector (Figure 6A,B). In addition, the overexpression of IGF2 resulted in the significant upregulation of *CD36*, *FATP4,* and *FABP5* in PTr2 cells (Figure 6C).

## 4. Discussion

Placental efficiency is closely associated with the growth and development of the fetus during pregnancy, which is determined by the exchange surface area, barrier thickness, blood flow, and expression of nutrient transport proteins [23]. The transport capacity of the placenta is believed to represent a key determinant of fetal nutrient availability and growth [24]. In this study, we found that the morphologies of placental folds tended to be complete at Day 65 of gestation; in particular, both the width of the folds and the number of trophoblast cells were significantly increased than those of the D40 placenta, which was consistent with the findings of a previous study [21]. The changes of the placental folds may correlate with the rapid growth of the developing pig fetus in the later stage of pregnancy. 

In the fetal growth trajectory, to grow the body during the middle and late trimesters, the fetus must obtain adequate nutrients from the maternal blood and this relies on the strengthening of the placental transport capacity of FAs. In this study, we observed that FATP4, FABP5, and CD36 had higher protein expression levels compared with other placental FA carriers on D40, D65, and D95 of gestation, implying that these proteins may be the main FA carriers in pig placenta. 

FA carriers in the placenta are responsible for the transport of fatty acids between fetal and maternal compartments, so alterations in the expressions of placental FA carriers can change the bulk of FAs across placental barriers. We observed that the expression levels of FATP4, FABP5, and CD36 had significantly upregulated trends in the placentae from D40 to D95 of gestation. Therefore, FATP4, FABP5, and CD36 may be responsible for the transport of adequate FAs that support fetal and placental growth and development. Studies in humans reported that the expressions of FATP4 and CD36 were directly correlated with the percentage of PUFAs in placentae [7,25]. FATP4 in human placenta is important in mediating the maternal–fetal docosahexaenoic acid (DHA) transfer [8,26], while FATP-4 deficiency in mice will result in early embryonic lethality [27]. CD36 is an integral membrane protein, and, thus, it can bind FAs, collagen, thrombospondin, and oxidized low-density lipoproteins. CD36 is also involved in FA uptake [28,29,30]. FABPs act in the cytoplasmatic compartment and are responsible for the intracellular distribution of FAs to determine their metabolic fate, including metabolism, transport, and membrane incorporation [31,32]. Alterations in FABPs in the placenta have been associated with several fetal complications, such as intrauterine growth restriction (IUGR) [28] and being small-for-gestational-age in humans [33], sheep [34], and mice [35].

The enrichment of LC-PUFAs in fetal circulation can be related to their interaction with several FA carriers, such as placental membrane FABP and FATP4 [36]. In vitro experimental studies and clinical cases of pregnant women studied by isotope labeling have demonstrated that a number of long-chain PUFAs (LC-PUFAs) from the dam facilitated by placental delivery, particularly AA and DHA (22:6n–3), are deposited in fetal tissues during the third trimester of pregnancy, when placental preferential transport of LC-PUFAs occurs in maternal circulation [8,25]. In this work, we observed that the full-term pig placenta had similar expression intensities of FATP4, FABP5, and CD36. This result further highlights the importance of these placental FA carriers for fetal growth and development in transporting adequate PUFAs from maternal circulation. The late intrauterine and early postnatal periods are characterized by the extensive deposition of these LC-PUFAs in offspring tissues [37].

Physiologically, before the second trimester, the placenta needs to acquire and store FAs, especially long-chain FAs (LCFAs), for cell proliferation, angiogenesis, and vascularization to achieve the integrity of placental morphologies and functions in preparation for the rapid fetal growth in late pregnancy [3]. In the present study, the levels of multiple long-chain FAs in porcine placentae dramatically increased from D40 to D65 of pregnancy, which corresponded to the increase in placental folds that maximize the interface between the fetal placenta and the endometrium. Furthermore, the development of a placental vascular network is fundamental for the growth and maintenance of the growing fetus. A recent study reported that long-chain FAs stimulated angiogenesis in placental trophoblast via vascular endothelium growth factor, angiopoietin-like protein 4, FABPs, and eicosanoids [5]. Inadequate placental angiogenesis results in structural and functional deficiency of the placenta, which may be responsible for several pregnancy complications, such as preeclampsia, pre-term IUGR, and spontaneous abortion [37]. In humans, the upregulation of the mRNA expressions of *LPL*, *FATPs* (-*1*, -*2*, and -*4*), and *FABPs* (-*1* and -*3*) was detected in IUGR placentae; this phenomenon may be a compensatory mechanism, but the IUGR fetus still failed to acquire a normal LC-PUFA supply [31]. Perazzolo et al. [25] applied a combined experimental and computational modeling approach to search for the incorporation of FAs into placental lipid pools, which may modulate their transfer to the fetus.

IGF2 and H19 are key imprinted genes that regulate the growth and functions of the placenta. We found that the mRNA expression level of *IGF2* in D95 placentae was significantly higher than that in D65 placentae. In mice, conditional IGF2 deletion was found to reduce the volume of the syncytiotrophoblast and glycogen trophoblast at junctions in the placenta of female fetuses, and to alter the expression of important endocrine and signal transduction-related genes [38]. IGF2 promotes mitosis, metabolism, differentiation, and survival of many cell types and its synthesis is mainly mediated by binding to IGF1R or insulin receptors to activate the PI3K-Akt or RAS-MAPK-ERK pathways [38]. Previous studies have shown that the epigenetic status of IGF2/H19 is unstable under the influence of the environment during pregnancy, including maternal factors and nutrient supply [39]. Therefore, upregulated expression levels of IGF2 in the placenta at late pregnancy may be an adaptation of the placenta in response to the rapid growth of the fetus. 

Furthermore, we found the overexpression of IGF2 promoted fatty acid uptake and resulted in significantly elevated expressions of *CD36*, *FATP4*, and *FABP5* in PTr2 cells. Much evidence from animal and clinical research has demonstrated that IGF2 plays an important role in lipid metabolism regulation [40,41]. A recent study in atherosclerosis demonstrated that the overexpression of IGF2 enhances oxidized low-density lipoprotein-induced CD36 expression, resulting in increased lipid accumulation in human THP-1 macrophages [42]. Therefore, placental IGF2 expression may have an impact on fetal fatty acid consumption by promoting the upregulation of placental FA transporters during late pregnancy. Further research is necessary, nevertheless, to determine how IGF2 interacts with FA transporters and other regulators of lipid metabolism in the porcine placenta.

In summary, we reported for the first time that CD36, FATP4, and FABP5 may be the main FA carriers in the pig placenta, and their expression levels exhibited significantly upregulated trends with the pregnancy progression, which may be associated with the dramatically increased levels of several important LCFAs in the placentae throughout pregnancy. Furthermore, the transcription level of *IGF2* in D95 placentae was dramatically upregulated, and in vitro experiments revealed that IGF2 enhances fatty acid uptake and the expression of *CD36*, *FATP4*, and *FABP5* in PTr2 cells. Therefore, FATP4, FABP5, and CD36 may be important regulators that enhance the transport of LCFAs in the pig placenta, supporting fetal and placental growth and development during the third trimester of pregnancy. Moreover, IGF2 may be involved in FA metabolism by affecting placental fatty acid transporters expression, which may be an adaptation of the placenta in response to the rapid growth of the fetus during late pregnancy.

## Figures and Tables

**Figure 1 genes-14-00872-f001:**
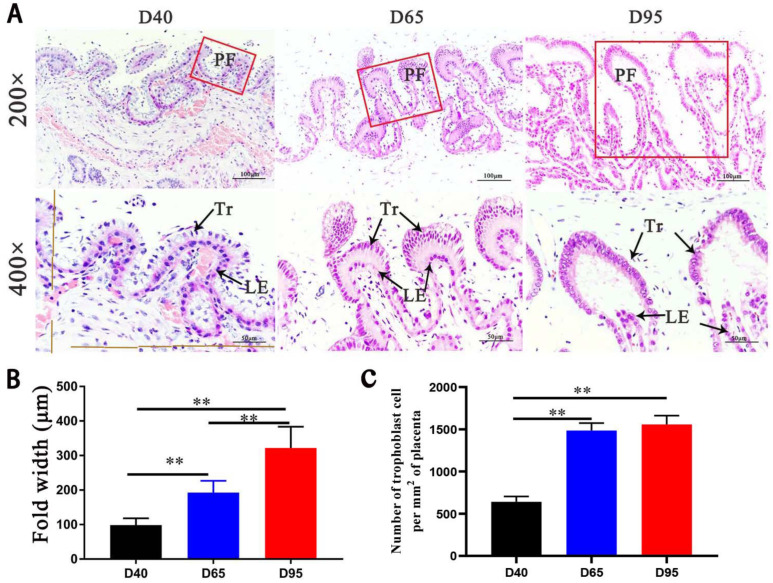
Comparison of placental microscopic folds at different stages of pregnancy. (**A**) Photomicrographs of representative maternal–fetal interface in pig pregnancies. Placental trophoblast (Tr) attached to endometrial luminal epithelium (LE) to form placental folds (PF). (**B**) The width of the PF and (**C**) the number of trophoblast cell per unit area of placenta were compared using morphometric measurements. ** *p* < 0.01.

**Figure 2 genes-14-00872-f002:**
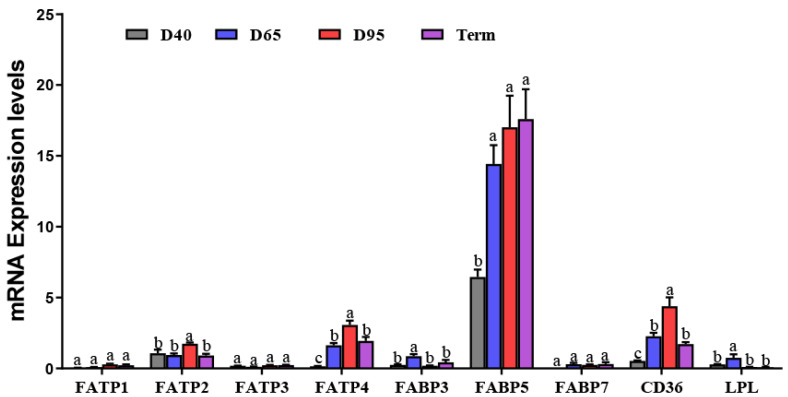
The mRNA expression levels of FA transport-related genes in porcine placentae at different gestation periods. Values are presented as mean ± standard error of the mean (SEM). Values labeled with different letters mean they are significantly different (*p* < 0.05) within the same gene group. Values labeled with the same letter mean they are not significantly different (*p* > 0.05) within the same gene group.

**Figure 3 genes-14-00872-f003:**
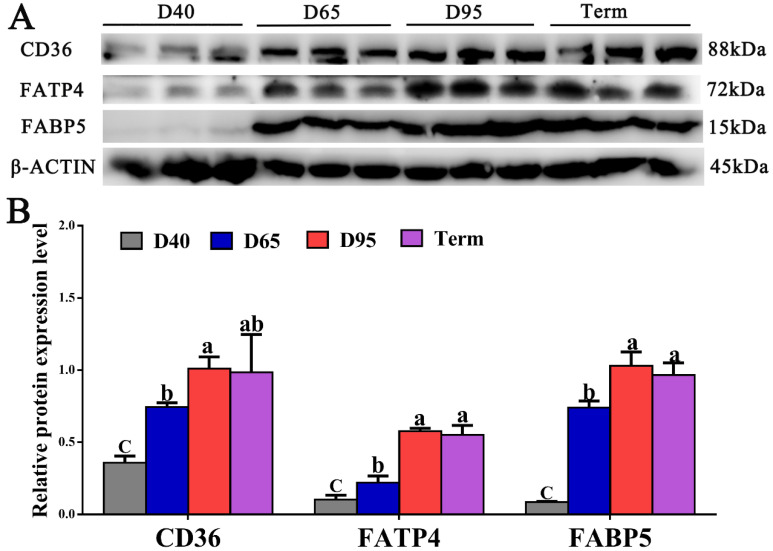
Comparison of protein expression levels of FA transport-related proteins in porcine placentae on D40, D65, D95, and term gestation. (**A**) Western blot analyses of protein expression levels and endogenous control β-actin. (**B**) Relative quantification of protein expression levels. Values are presented as mean ± SEM. Values labeled with different letters mean they are significantly different (*p* < 0.05) within the same gene group. Values labeled with the same letter mean they are not significantly different (*p* > 0.05) within the same protein group.

**Figure 4 genes-14-00872-f004:**
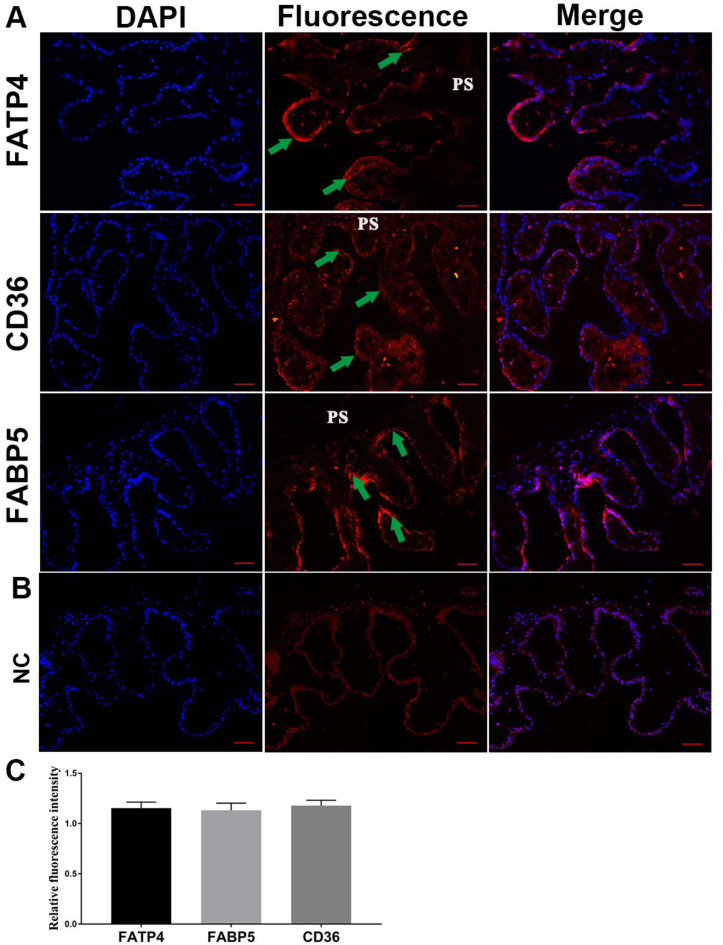
Localization of FATP4, CD36, and FABP5 by IF in pig term placentae. (**A**) Left column: nuclear staining with 4′,6-Diamidino-2-Phenylindole (DAPI); middle column, localization, and expression of FATP4, CD36, and FABP5 detected by IF staining; right column, merged images of DAPI and fluorescence. (**B**) Negative control (NC), where normal mouse IgG were substituted for primary antibody. (**C**) Relative quantification of fluorescence intensity of FATP4, CD36, and FABP5 protein in full-term pig placentae. PS, placental stroma; green arrow, cytotrophoblastic layer. Scale bar = 100 µm.

**Figure 5 genes-14-00872-f005:**
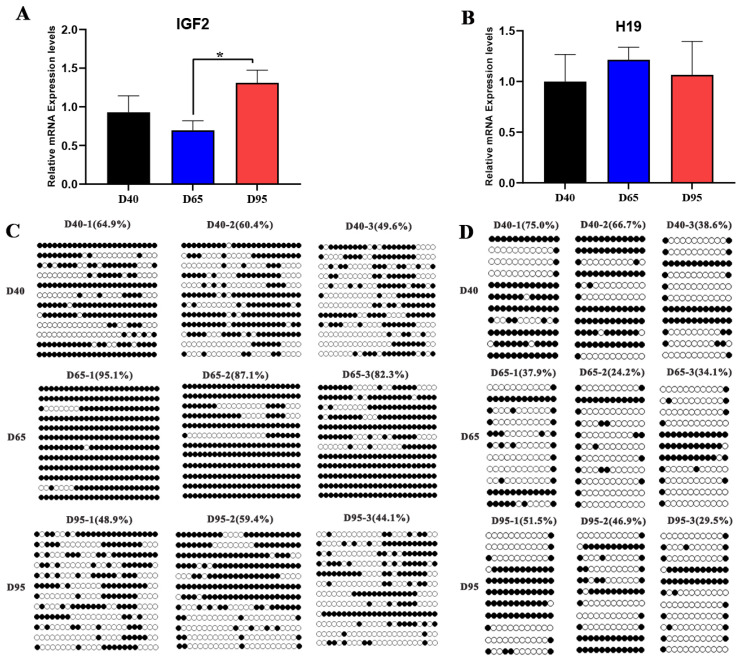
The expression levels and methylation levels of *IGF2* and *H19* in pig placentae across different gestation periods. The relative mRNA expression levels of *IGF2* (**A**) and *H19* (**B**) in pig placentae across different gestation periods. Methylation level of *IGF2* DMR2 (**C**) and *H19* CTCF3 (**D**) in pig placentae on D40, D65, and D95 of gestation. Values labeled with an asterisk (*) mean they are significantly different (*p* < 0.05) within the same gene group. Black and white circles represent methylated and unmethylated CpGs, respectively.

**Figure 6 genes-14-00872-f006:**
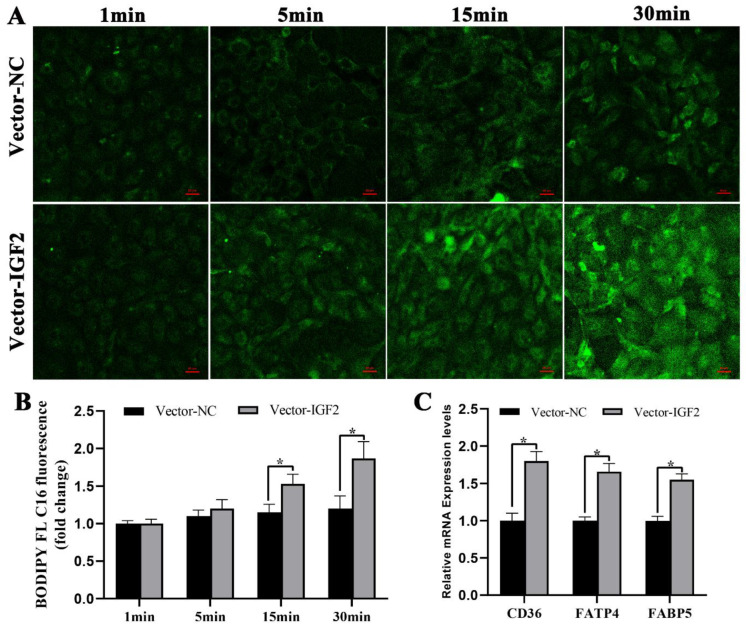
Overexpression of IGF2 enhances fatty acid uptake by PTr2 cells. (**A**) Representative photographic of uptake of BODIPY FL C16 by PTr2 cells after transfection with IGF2 overexpression vector. Scale bar = 50 μm. (**B**) The quantification of BODIPY FL C16 fluorescence. (**C**) The effect of IGF2 overexpression on mRNA expression levels of *CD36*, *FATP4,* and *FABP5* in PTr2 cells. Values are presented as mean ± S.E.M. Values labelled with asterisk (*) mean they are significantly different (*p* < 0.05).

**Table 1 genes-14-00872-t001:** Primers used for qPCR.

Genes	Primer Sequences (5′ to 3′)	Product Size (bp)	Amplification Efficiency	GenBank Accession No.
*FATP1*	F:GTGCTGGTGATGGACGAACTGG	180	98.76%	XM_021076151
	R:GCCTGCTTTGCCCTCTACTCCT			
*FATP2*	F:GGACGAGACGCTCACCTATG	185	102.04%	NM_001278777
	R:TGTAGTTGAGGCACGCCATG			
*FATP3*	F:AGGTCTCAGCCGAAGTGGATGC	127	104.21%	XM_021089805
	R:TGACTGATCCGAGCAGCCTTGG			
*FATP4*	F:TATGGTGTGGAGGTGCCAGGAA	198	103.49%	XM_013993903
	R:CCGCAGGTCTGTCTTCTGTAGC			
*FABP3*	F:AGCACCTTCAAGAGCACAGAGA	105	101.65%	NM_001099931
	R:GCCTCCATCCAGTGTCACAATG			
*FABP5*	F:GGCAAAGACCTCACCATCAAA	182	96.13%	NM_001039746
	R:CTTGTGATTGTGCTCTCCTTCC			
*FABP7*	F:AGGCAGGTGGGAAATGTGACT	300	95.45%	NM_001025229
	R:CTCATAGTGGCGAACAGCAACC			
*CD36*	F:AGGACCCTGAGACCCACACA	116	97.34%	NM_001044622
	R:TGCCACAGCCAGATTGAGAACA			
*LPL*	F:CATTGTGGTGGACTGGCTGTCT	290	95.12%	NM_214286
	R:AATCTGCGTCGTCGGGAGAAAG			
*IGF2*	F:CCCCCCTTCCTTCTCTTTCTT	101	-	[18]
	R:GCGACAAGCCTACCTGCAA			
*H19*	F:GGCCGGAGAATGGGAAAGAAGG	148	-	[18]
	R:CGCAGTGCTGCGTGGGAACG			
*β-Actin*	F:CCACGAGACCACCTTCAACTC	131	96.56%	DQ845171
	R:TGATCTCCTTCTGCATCCTGT			

**Table 2 genes-14-00872-t002:** Comparison of the fatty acid concentrations in the placentae from different gestation periods.

Detected Fatty Acids	D40 (n = 8)	D65 (n = 8)	D95 (n = 8)	Term (n = 8)
Caproic acid (C6:0)	1.08 ± 0.03 ^a^	1.04 ± 0.06 ^ab^	1.03 ± 0.09 ^ab^	0.96 ± 0.03 ^b^
Caprylic acid (C8:0)	1.54 ± 0.37	1.06 ± 0.19	1.47 ± 0.42	1.07 ± 0.25
Capric acid (C10:0)	0.96 ± 0.04 ^a^	0.94 ± 0.05 ^ab^	0.94 ± 0.1 ^ab^	0.84 ± 0.07 ^b^
Lauric acid (C12:0)	1.98 ± 0.11	1.92 ± 0.2	1.9 ± 0.22	1.81 ± 0.11
Tridecanoic acid (C13:0)	7.04 ± 0.31	7.65 ± 0.81	7.62 ± 1.07	8.12 ± 0.59
Myristic acid (C14:0)	22.09 ± 1.93	25.62 ± 4.48	25.95 ± 1.1	22.57 ± 2.21
Myristelaidic acid (C14:1T)	194.55 ± 17.7 ^a^	161.33 ± 22.08 ^b^	143.1 ± 4.34 ^bc^	126.11 ± 9.67 ^c^
Myristoleic acid (C14:1)	42.26 ± 5.18 ^a^	36.97 ± 3.71 ^b^	33.63 ± 1.28 ^b^	35.27 ± 1.27 ^b^
Pentadecanoic acid (C15:0)	13.1 ± 0.36 ^a^	13.8 ± 4.87 ^a^	14.31 ± 3.01 ^a^	5.9 ± 0.58 ^b^
*Trans*-10-Pentadecenoic acid (C15:1T)	41.81 ± 3.05	41.19 ± 4.34	40.07 ± 1.52	43.79 ± 1.95
cis-10-Pentadecenoic acid (C15:1)	30.61 ± 3.25	30.76 ± 4.03	29.02 ± 1.22	31.16 ± 1.2
Palmitic acid (C16:0)	1889.42 ± 233.89	1894.75 ± 190.05	2049.28 ± 176.39	1847.35 ± 139.56
*Trans*-Palmitelaidic acid (C16:1T)	77.96 ± 6.74	75.84 ± 9.04	72.64 ± 3.84	78.39 ± 3.07
Palmitoleic acid (C16:1)	65.69 ± 7.58	85.86 ± 18.33	92.68 ± 31.75	64.61 ± 3.07
Heptadecanoic acid (C17:0)	25.71 ± 3.06 ^a^	21.16 ± 7.41 ^a^	24.03 ± 6.34 ^a^	12.11 ± 0.69 ^b^
Methyl Trans-10-Heptadecenoic Acid (C17:1T)	28.55 ± 2.06	27.76 ± 2.61	26.17 ± 1.04	27.86 ± 1.15
10-Heptadecanoic acid(C17:1)	44.13 ± 5.51 ^a^	39.47 ± 3.79 ^ab^	38.06 ± 2.39 ^b^	36.67 ± 1.32 ^b^
Stearic acid(C18:0)	1172.27 ± 122.3	1122.27 ± 137.76	1170.75 ± 117.48	1016.42 ± 112.09
Petroselaidate (C18:1N12T)	20.76 ± 1.38	20.98 ± 1.87	20.1 ± 1	19.96 ± 0.72
Elaidate (C18:1N9T)	8.72 ± 0.59	9.16 ± 0.86	9.04 ± 0.46	9.29 ± 0.29
Transvaccenate (C18:1N7T)	69.9 ± 3.92	69.92 ± 7.09	67.77 ± 1.96	68.73 ± 2.54
Petroselinic acid (C18:1N12)	126.95 ± 30.07 ^b^	154.43 ± 24.74 ^b^	220.44 ± 59.18 ^a^	164.89 ± 43.03 ^ab^
Oleic acid (C18:1N9C)	295.12 ± 141.51 ^d^	749 ± 30.53 ^b^	1074.14 ± 44 ^a^	495.38 ± 67.89 ^c^
Vaccenic acid (C18:1N7)	111.01 ± 42.52 ^c^	246.83 ± 24.95 ^b^	352.68 ± 16.59 ^a^	117.76 ± 8.58 ^c^
Linoelaidate (C18:2N6T)	7.28 ± 0.45	7.77 ± 0.99	8.32 ± 1.26	7.61 ± 0.3
7-Transnonadecenoate (C19:1N12T)	16.71 ± 0.71	17.62 ± 1.89	17.26 ± 1.33	17 ± 0.95
10-Transnonadecenoate (C19:1N9T)	0.64 ± 0.08 ^a^	0.52 ± 0.08 ^ab^	0.57 ± 0.12 ^ab^	0.47 ± 0.05 ^b^
Linoleic acid (C18:2N6)	24.67 ± 11.69 ^d^	79.63 ± 7.54 ^b^	116.56 ± 8.69 ^a^	59.14 ± 5.12 ^c^
Arachidic acid (C20:0)	9.97 ± 0.6	10.57 ± 1.98	11.4 ± 0.49	10 ± 0.82
γ-Linoleic acid (C18:3N6)	2.96 ± 0.32	3.48 ± 0.51	3.4 ± 0.39	3.06 ± 0.18
Eicosenic-*cis*-11 acid (C20:1T)	45.39 ± 3.24	45.83 ± 4.74	44.58 ± 2.21	46.55 ± 2.17
Eicosenic-cis-5 acid (C20:1)	29 ± 5.24	30.32 ± 4.09	30.42 ± 3.2	27.86 ± 0.84
α-Linoleic acid (C18:3N3)	1.99 ± 0.13	2.09 ± 0.22	2.08 ± 0.3	2.41 ± 0.32
*cis*-11,14-Eicosadienoate acid (C20:2)	8.39 ± 2.45	10.58 ± 2.07	11.2 ± 2.38	8.4 ± 0.26
Behenic acid (C22:0)	1.51 ± 0.14 ^b^	3.81 ± 2.53 ^ab^	4.2 ± 1.44 ^a^	3.14 ± 0.3 ^ab^
cis-8,11,14-Eicosadienoate acid (C20:3N6)	8.96 ± 3.31 ^c^	19.6 ± 6.3 ^ab^	21.95 ± 8.14 ^a^	12.14 ± 1.65 ^bc^
Brassidate (C22:1N9T)	23.01 ± 1.23	22.68 ± 2.61	22.99 ± 2.13	22.5 ± 3.26
Erucate (C22:1N9)	22.15 ± 1.2	22.74 ± 1.65	21.72 ± 1.19	22.29 ± 0.91
*cis*-11,14,17-Eicosadienoate acid (C20:3N3)	1.86 ± 0.16	2.24 ± 0.41	2.3 ± 0.32	2.16 ± 0.02
Arachidonic acid (C20:4N6)	25.48 ± 5.68 ^b^	43.03 ± 13.51 ^ab^	50.23 ± 22.33 ^a^	36.75 ± 5.39 ^ab^
Tricosanoic acid (C23:0)	0.57 ± 0.1	0.78 ± 0.45	0.83 ± 0.26	0.57 ± 0.06
Docosanoic acid (C22:2)	5.38 ± 0.69	5.67 ± 0.18	5.48 ± 0.33	5.56 ± 0.22
Eicosapentaenoic acid (C20:5N3)	2.2 ± 0.08 ^b^	3.1 ± 0.32 ^a^	3.25 ± 0.89 ^a^	3.61 ± 0.35 ^a^
Lignoceric acid (C24:0)	1.77 ± 0.24 ^b^	8.32 ± 4.64 ^a^	9.54 ± 4.56 ^a^	6.39 ± 0.76 ^a^
Nervonic acid (C24:1)	22.97 ± 5.11	26.89 ± 4.96	27.86 ± 1.76	26.64 ± 1.72
Docosatetraenoic acid (C22:4)	12.57 ± 1.3 ^b^	26.2 ± 12.31 ^a^	22.68 ± 6.84 ^ab^	20.45 ± 2.25 ^ab^
Docosapentaenoic acid (n-6) (C22:5N6)	16.92 ± 9.41	26.53 ± 6.1	31.59 ± 14.01	20.46 ± 2.89
Docosapentaenoic acid (n-3) (C22:5N3)	5.63 ± 1.65 ^b^	9.06 ± 1.43 ^a^	9.63 ± 2.2 ^a^	9.34 ± 0.61 ^a^
Docosahexaenoic acid (C22:6N3)	9.9 ± 2.91 ^b^	15.68 ± 3.68 ^a^	17.97 ± 6.27 ^a^	13.1 ± 1.37 ^ab^

Data are expressed as mean ± SD (μg/g) and one-way analysis of variance (ANOVA) was performed to evaluate group differences. Values labelled with different letters mean they are significantly different (*p* < 0.05) within the same row (*p* < 0.05). FA, Fatty acid.

## Data Availability

The data that support the findings of this study are available from the corresponding author.

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
