# Peer review of "IGF2 May Enhance Placental Fatty Acid Metabolism by Regulating Expression of Fatty Acid Carriers in the Growth of Fetus and Placenta during Late Pregnancy in Pigs"

_genes, 2023, doi:10.3390/genes14040872_

Round 1
Reviewer 1 Report
The abstract as well as the discussion sections needs to be rewritten. Please see the attached edited pdf

Author Response
Dear Reviewer:
Thank you very much for reviewing our manuscript and giving your constructive comments. We have revised the abstract and the discussionour manuscript according to your comments.
Reviewer 2 Report
General Summary and Comments: The authors note that fatty acids (FAs) are essential for growth and development of the fetus and the placenta and that they are derived from maternal blood by various placental FA carriers, including FA transport proteins (FATPs), FA 13 translocase (FAT/CD36), cytoplasmic FA binding proteins (FABPs). They also indicate H19 and IGF2 play crucial roles in placental growth and functions, although the relationship between expression patterns of H19/IGF2 and placental fatty acid metabolism throughout pig pregnancy is unclear. Thus, the authors report results of a study in which they investigated fatty acid profiles, expression patterns of FA 17carriers and H19/IGF2 in placentae from sows on Days 40 (D40), 65 (D65) and 95 (D95) of pregnancy. They report the following: 1) placental folding tended to be complete at D65, especially the width of placental folds and the number of trophoblast cells increased significantly; 2) important long-chain FAs 20 (LCFAs), including oleic acid, linoleic acid, arachidonatic acid, eicosapentaenoic acid, docosatetraenoatic acid, increased in the placentae throughout pregnancy; 3) expression of CD36, FATP4, and FABP5 is greater compared with other FA 23 carriers with expression increasing between Days 40 and 95 of pregnancy; 4) increases in expression of IGF2 mRNA was associated with less DNA methylation in DMR2 of IGF2 in placentae from Day 95 as compared to those from Day 65; and 5) over-expression of IGF2 significantly increased uptake of fatty acids and expression of CD36, FATP4 and FABP5 in PTr2 cells. The authors conclude that CD36, FATP4, and FABP5 may be important regulators of transport of long chain fatty acids and that IGF2 be involved in FA metabolism by affecting expression of FA carriers to support the growth of fetuses and placentae during pregnancy in pigs.
The authors have not described the experimental design sufficiently or analyzed the data in sufficient detail to allow one to fully assess the significance of their results. One major flaw it the lack of a negative control for IF.
The manuscript requires considerable editing for grammar.
Specific Comments/Questions:
There is an additional and perhaps better reference regarding placental folding in pigs. Please see the following reference. Mechanotransduction drives morphogenesis to develop folding during placental development in pigs. Seo H, Li X, Wu G, Bazer FW, Burghardt RC, Bayless KJ, Johnson GA. Placenta. 2020 Jan 15;90:62-70.
Lines 56-59 – How do fatty acids contribute to cell proliferation? Are they mitogens or are they only required to synthesize lipid membranes?
Lines 60-70 – The authors have not provided a clear rationale for linking H10/IGF2 to metabolism and transport of fatty acids. Why did that choose to study placentae from Days 40, 65 and 95 of gestation, as well as term placentae?
Section 2.1 – The authors state that sows were inseminated thrice, but do not state, relative to the detection of estrus, when those inseminations were performed. What is meant by “uteri were opened from the corners.” Uteri of pigs do not have corners, so this statement needs clarification. How many placentae were collected and analyzed for each of the sows on each of the selected days of gestation. Later the authors state that they analyzed placentae using qPCR, but there is no mention of snap freezing samples of placentae or how many placentae from each sow were analyzed for expression of selected mRNAs using qPCR analyses, western blotting and bisulfite sequencing analyses?
Section 2.7 – For IF, what was the negative control used to account for nonspecific staining? The absence of the primary antibody is required. The use of PBS is not a control, so the results of IF are very questionable. The proper control requires use of an irrelevant antibody at the same concentration as the primary antibody to account for nonspecific staining.
Line 195 – What are “competent DH5α?
Section 2.9 and 2.10 – What is the established phenotype of the PTr2 cells? How were they validated to establish that their phenotype recapitulates in vivo characteristics of trophoblast/chorion of pigs? Why didn’t the authors culture sections of choriallantois to determine uptake of fatty acids? Please provide more details regarding the BODIPY fatty acids rather than just stating that they were very long chain fatty acid analogues.
Lines 218-220 – the sentence does not make sense as written.
Section 2.11 – The experimental design is not provided. Did the analyses determine main effects of Day of Gestation for the variables measured. The authors should use least squares regression analyses to determine if changes in variables were linear, curvilinear and so forth. What the authors describe is not a rigorous analysis of data.
Line 231 – Do you mean 45 piglets delivered from five sows at farrowing? There is no mention of this group of sows previously, so how were placental tissues obtained for analyses?
Lines 233-238 – The authors must provide quantitative data to justify the statements on numbers of trophoblast cells as they have done for widths of placental folds. Figures 1A and 1 B do not provide any information on numbers of trophoblast cells. Analyses of data using least squares regression analyses would likely confirm that widths of placental folds increase linearly between Days 40 and 95 of gestation.
Table 2: The title should be Concentrations of fatty acids in placentae from Days 40, 65 and 95 of gestation and at term. Question: How were the 8 placentae selected at each of the stages of gestation and at term? That has not been explained. Why were only 8 placentae from each day evaluated?
Figure 2 legend: How many placentae were subjected to qPCR analyses?
Figure 3 legend: How many placentae were subjected to western blotting. Are the data based on analyses of only 3 placentae from each day of gestation.
Figure 4: There is no negative control and no indication of how many placentae were analyzed usijng IF analysis.
Figure 5: How many placentae were analyzed?
Figure 6: How many experiments were there and how many replicates for each time point (Panel B) and the same question for results in Panel C.
There is a concern overall that the authors collected placentae and pooled them to for their analyses. If so, depending on how the placental samples were pooled, there may be only an N=1 or N=2 per day of gestation. The experimental design must be clarified in detail.
Reviewer 3 Report
Paper with original contributions in genetics and reproduction in an animal species of biological and productive importance. Mining suggestion regarding the reformulation of the title of the article in the attached archive. Best regard

Author Response
Dear Reviewer:
Thank you very much for reviewing our manuscript and giving your constructive comments. We have revised our manuscript according to your comments.
Round 2
Reviewer 2 Report
This revised manuscript requires considerable additional editing. I do not accept PBS as a negative control for IF, so the results in Figure 4 are not acceptable.
Author Response
Dear Reviewer:
Thank you very much for giving your constructive comments. We have added the negative controls in figure 4 in which normal mouse IgG at the same concentration was used as the primary antibody according to the methods as described previously (Seo et al. Placenta. 2020,90:62-70). Thanks for your recommendation.
In addition, our updated manuscript underwent first round of English editing (English editing ID: English-62616) through a service that the genes journal recommended. In accordance with your advice, we had a second round of English editing done by a reputable editing service ShineWrite (www.ShineWrite.com), and we also invited a peer expert to revise our manuscript.